

# Improving smart home surveillance through YOLO model with transfer learning and quantization for enhanced accuracy and efficiency

Surjeet Dalal[1], Umesh Kumar Lilhore[2], Nidhi Sharma[3], Shakti Arora[4], Sarita Simaiya[2], Manel Ayadi[5], Nouf Abdullah Almujally[5] and Amel Ksibi[5]

[1] Department of Computer Science and Engineering, Amity University Haryana, Manesar, Haryana, India
[2] Department of Computer Science and Engineering, Chandigarh University, Mohali, Punjab, India
[3] Computer Applications, GLA University, Mathura, UP, India
[4] Department of Computer Science and Engineering, PIET Panipat, Panipat, Haryana
[5] Department of Information Systems, College of Computer and Information Sciences, Princess Nourah bint Abdulrahman University, Riyadh, Saudi Arabia

Corresponding author
Nouf Abdullah Almujally,
naalmujally@pnu.edu.sa

## ABSTRACT

The use of closed-circuit television (CCTV) systems is widespread in all areas where serious safety concerns exist. Keeping an eye on things manually sounds like a time-consuming and challenging process. Identifying theft, detecting aggression, detecting explosive risks, *etc.*, are all circumstances in which the term "security" takes on multiple meanings. When applied to crowded public spaces, the phrase "security" encompasses nearly every conceivable kind of abnormality. Detecting violent behaviour among them is challenging since it typically occurs in a group setting. Several practical limitations make it hard, though complex functional limitations make it difficult to analyze crowd film scenes for anomalous or aberrant behaviour. This article provides a broad overview of the field, starting with object identification and moving on to action recognition, crowd analysis, and violence detection in a crowd setting. By combining you only look once (YOLO) with transfer learning, the model may acquire new skills from various sources. This makes it more flexible for use in various object identification applications and lessens the time and effort required to gather large annotated datasets. This article proposes the YOLO model with transfer learning for intelligent surveillance in Internet of Thing (IoT)-enabled home environments in smart cities. Quantization concepts are being applied to optimize the YOLO model in this work. Using YOLO with quantization, the model is optimized for use on edge devices and mobile platforms, which have limited computing capabilities. Thus, even with limited technology, object detection systems may be used in various real-world applications. The proposed model has been validated on two different datasets of 7,382 images. The proposed model gains an accuracy level of 98.27%. The proposed method outperforms the conventional one. The use of the YOLO model and transfer learning with quantization has significant potential for enhancing ecological smart city monitoring, and further research and development in this area could contribute to developing more effective and efficient environmental smart city monitoring systems.

# INTRODUCTION

Smart city surveillance uses advanced technologies such as cameras, sensors, and data analysis tools to monitor public spaces and improve safety, security, and efficiency in urban areas. These technologies can provide real-time data on traffic patterns, public transportation usage, environmental conditions, and even potential crimes or emergencies. This data can then be analysed to identify trends, make predictions, and inform resource allocation and policy development decisions. While intelligent city surveillance can provide numerous benefits, such as reducing crime and traffic congestion, it also raises concerns about privacy, civil liberties, and the potential for misuse or abuse of data. Therefore, cities must establish clear guidelines and regulations to ensure these technologies are used responsibly and ethically, with proper safeguards to protect individual rights and freedoms. The concept of "smart home security" refers to the usage of various Internet of Things (IoT) enabled gadgets to enable consumers to keep tabs on and control the safety of their houses from afar. If smart locks are installed, these systems may control who enters and leaves a property and manage the security of its exterior (*Kumar et al., 2019*). Forecasts place the value of the IoT security industry at $35.21 billion in 2023, up from $8.2 billion in 2018. Smart home security systems allow customers to keep tabs on their dwellings from afar and receive instant notifications about suspicious activity or unauthorized entry attempts. Although typical home alarm systems stop monitoring and sending alerts when the alarm is silenced, intelligent security systems continue to do so. Smart home security systems can feature motion-activated, remotely accessible security cameras and motion-detection doorbells that can be used to identify and interact with an intruder before unlocking the door (*Şerban et al., 2019*).

That the IoT improves our quality of life at home is undeniable. Moreover, the users are capable to manage the temperature, turn off the lights, secure the doors, and monitor the home remotely from the mobile device. The IoT is a system that links together numerous electronic gadgets to form a single network. As the installed devices can be accessed through the network, the users have complete control over them wherever it happen. Lights, air conditioners, fridges, and fans have energy efficiency settings that have been adjusted to save costs and carbon emissions. Connected devices to the Internet make our daily lives easier, safer, and more comfortable (*Erzi & Aydin, 2020*).

## Problem formulation

IoT devices can monitor who enters and exits the residence if smart locks are installed in the doors. Relax knowing that the home is safe with the help of a high-tech security system. For instance, a smart doorbell can recognize a stranger, initiate communication, and decide whether to unlock the gate. The high-definition cameras in the gadgets are triggered by motion detection. In the event of an intrusion, these systems alert the users, set the alarm, and contact the authorities. Having a state-of-the-art security system for the house have provided peace of mind and many other advantages. The Internet of Things allows user to do amazing things, like control the home from miles away with an app on the phone. Despite the potential benefits of smart city surveillance, several research gaps

must be addressed to ensure these technologies are used effectively and responsibly (*Huang et al., 2020*). There are some challenges faced in this domain:

- *Privacy and security concerns:* One of the main concerns with smart city surveillance is the potential for privacy violations and security breaches. Further research is needed to understand the specific risks and develop effective strategies to address them.
- *Ethical and legal issues:* There are ethical and legal issues that arise from the use of smart city surveillance, including questions about consent, transparency, and accountability. Further research is needed to identify and address these issues.
- *Impact on social inequality:* There is a concern that smart city surveillance could exacerbate social inequality by disproportionately affecting certain groups of people. Research is needed to understand the potential impact on different communities and develop strategies to mitigate any adverse effects.
- *Data quality and reliability*: Smart city surveillance relies on accurate and reliable data. However, there may be issues with data quality, such as bias or incomplete data. Further research is needed to develop methods for ensuring data quality and reliability.
- *Public perception and acceptance*: The success of smart city surveillance depends on the public's perception and acceptance of these technologies. Further research is needed to understand public attitudes towards smart city surveillance and identify factors influencing acceptance.
- *Integration with other technologies*: Smart city surveillance is just one component of a larger ecosystem of smart city technologies. Further research is needed to understand how different technologies can be integrated to create a cohesive and influential smart city system.

Addressing these research gaps will ensure smart city surveillance is implemented responsibly and effectively (*Mahamuni & Jalauddin, 2021*).

## Motivation and critical contribution of the research

Every day, more theft occurs, resulting in monetary and non-monetary losses. The following research motivates the authors to carry out this research work.

*Research questions:*

**RQ1:** Can the you only look once (YOLO) model be improved using transfer learning, especially for recognizing typical household objects in a smart home surveillance setting?

The goal of the research question is to find out if the YOLO model can be trained with transfer learning on a varied dataset of domestic goods to enhance its detection and identification of everyday things found in a smart home, including appliances, furniture, and personal possessions.

**RQ2:** Can the YOLO model be more effective for use in a resource-constrained smart home monitoring system by employing quantization techniques?

This research question probes the potential of utilizing quantization techniques to scale down the YOLO model's storage and processing needs without sacrificing precision. Gains in efficiency for real-time monitoring in a smart home setting are the primary focus. As a result, it is critical to recognize these threats and maintain system security

(*Vamsi et al., 2021*). This research uses deep learning algorithms to create a system detecting infiltrators in smart homes.

- This work introduces a smart home hybrid infiltrator detection model utilizing an improved YOLO model.
- Transfer learning is used to evaluate the hybrid and YOLO models on two prominent RoboFlow datasets.
- Quantization is used to optimize the YOLO model's parameters, which reduces the precision of the weights and activations, improving performance and memory footprint without losing accuracy.
- An experimental study was done utilizing precision, F-measure, recall, accuracy, and detection rate to compare the current and suggested models.

The model had 98.27% accuracy on Robolflow datasets. The complete research article is divided into several categories. 'Review of literature' will look at what's already written on spotting an intruder in a smart home. The third section describes the resources used for the current study and the recommended architecture and data sets. The findings and analysis of the experiments are presented in 'Results and Analysis'. The Conclusion discusses the last thoughts and what comes next.

## REVIEW OF LITERATURE

*Liu et al. (2017)* detail a method that uses deep neural networks and balanced sampling. The proposed procedure involves two basic measures. Using data augmentation by proportional sampling is the first step in addressing the issue of unequal data collection.

*Wang et al. (2018)* present a system based on CNN and transfer learning to enhance deep learning approaches for vehicle-type categorization from surveillance images. Both vehicle manufacturers and the Internet have many pictures of various cars with their labels on them. That's why our suggested vehicle type recognition system for surveillance applications relies entirely on information from the Internet to supply its labels. This model saves us time and effort during training by automatically categorizing information collected from surveillance images.

*Kaşkavalci & Gören (2019)* provide a deep learning-based, decentralized, scalable surveillance architecture using Edge and Cloud computing. The prevalence of innovative surveillance technologies is growing as they become more widely available and reasonably priced. The conventional surveillance technique involves constantly recording footage into a recording medium. However, doing so produces a vast amount of data and reduces the lifespan of a hard disk. The cloud is where video from today's cameras is stored. Using this feature will increase the monthly cost of the Cloud subscription since more bandwidth will be required. Our technology drastically reduces bandwidth and Cloud fees by processing film locally before uploading it.

When talking about live video monitoring, to get around problems with anomaly detection and localization, *Nawaratne et al. (2020)* propose the incremental spatiotemporal learner (ISTL). An unsupervised deep learning technique, ISTL combines active learning and fuzzy aggregation to detect and classify growing abnormalities over time.

For a specific application of IIoT, *Zhao, Yin & Gui (2020)* describe a deep learning-based INES approach. Their first contribution is a depthwise separable convolutional method for building a lightweight deep neural network with reduced processing requirements. Second, they utilize a hybrid computing strategy, integrating edge and cloud computing to minimize network stress. Finally, they employed their proposed INES method on an actual building site to validate an IIoT application. The proposed INES in the edge device can detect 16 frames per second. By combining edge and cloud computing, detection accuracy may be raised to as much as 89%. In addition, an edge device's operational expenses are one-tenth those of a central server. The proposed INES method is experimentally validated concerning computing cost and detection accuracy.

*Abdelali et al. (2021)* present a fully automated technique for monitoring traffic cameras called multiple hypothesis detection and tracking (MHDT). The proposed method can recognize and track cars throughout their journey despite obstacles, including vehicle size changes, stops, rotations, illumination changes, and occlusions.

*Ahmed et al. (2021)* present a system that uses a sophisticated robotic camera connected to a VPU. For this purpose, they have used two pre-trained deep learning models, SSD and YOLO. By comparing the findings across algorithms, they discovered that the accuracy of various tracking methods is relatively near to one another, sitting at 90% to 94%. Recommendations for the future and the outcomes have also been considered.

*Jung, Kang & Chun (2021)* use information gathered from Internet of Things gadgets to investigate the efficacy of anomaly analysis using deep Learning for facility management. They installed three environmental monitoring sensors in the building's cloakrooms and common spaces utilizing the Internet of Things (IoT). Each sensor collected around 53,000 time-series data recordings in about a month for training. With a MAPE of 8.58%, the accuracy of the predictions was high. The trained model may be used in anomaly detection by defining error thresholds.

A deep model that incorporates multi-scale deep supervision and attention feature learning is proposed by *Wu et al. (2021)* for PReID. The attention module can cause feature information to be lost; therefore, they built a multi-scale feature learning layer that is tightly supervised to train the network and prevent this from happening. The proposed modules are not used during testing but instead discarded in favour of newer versions.

When applied to security footage, the new model proposed by *Sivachandiran, Jagan Mohan & Mohammed Nazer (2022)* (DLD-APDT) uses deep learning (DL) to recognize and track people automatically. The primary goal of the proposed DLD-APDT model is to locate and track moving people inside video clips. One way the proposed DLD-APDT model does this is by a process known as frame conversion, in which the input video is converted into individual still images. Computer simulations demonstrated that the DLD-APDT technique outperformed state-of-the-art technologies in both detection and tracking. *Kulurkar et al. (2023)* present a novel IoT-based system that uses low-power wireless sensor networks, big data, cloud computing, and smart devices to detect indoor falls in older people. To monitor the movement of more senior citizens in real-time, they employed a wearable six LowPAN device fitted with a three-axis accelerometer. Using a machine learning model to analyze and evaluate sensor information on an advanced IoT

**Table 1  Summary of existing work.**

| S. No. | Article | Method | Dataset | Result |
|---|---|---|---|---|
| 1 | *Liu et al. (2017)* | Deep neural networks | MIO-TCD classification challenge dataset | 0.8844 Recall, 0.9776 Precison, |
| 2 | *Wang et al. (2018)* | CNN and transfer learning | Comprehensive cars (Comp-Cars) dataset | 55.13% Accuracy |
| 3 | *Kaşkavalci & Gören (2019)* | Deep learning | Google photos | 93.56% Accuracy |
| 4 | *Nawaratne et al. (2020)* | Incremental spatiotemporal learner (ISTL) | University of California San Diego (UCSD) Pedestrian datasets | 91.1 AUR |
| 5 | *Zhao, Yin & Gui (2020)* | ReLU6 deep learning model | Datasets of pedestrians and helmets | 0.68 mAP |
| 6 | *Abdelali et al. (2021)* | YOLO detection approach | MoVITS dataset | 92.50% Accuracy |
| 7 | *Ahmed et al. (2021)* | YOLO model | COCO dataset | 94% Accuracy |
| 8 | *Jung, Kang & Chun (2021)* | Long-short term memory (LSTM) model | Testbed | 8.58 MAPE |
| 9 | *Wu et al. (2021)* | ResNet-50 model | PReID datasets | 94% Accuracy |
| 10 | *Sivachandiran, Jagan Mohan & Mohammed Nazer (2022)* | EfficientDet model | Pascal VOC and PenFudan dataset | 92.95 Precision |
| 11 | *Kulurkar et al. (2023)* | LSTM model | MobiAct dataset | 95.87% Accuracy |
| 12 | Proposed model | YOLO model with transfer learning | 3,469 images dataset 1 8,324 images dataset 2 | 98.87% Accuracy |

gateway allows for very accurate fall detection. With their edge computing solution, user can detect falls with an accuracy of 95.87% using analytics performed in real-time on data streams.

To foresee the potential range of increase in COVID-19-infected patients in the coming days, *Cai et al. (2023)* present a deep-learning approach. This novel method is used in the proposed model to comprehend the interplay between several factors, such as population size, mobility within and between counties, social distance, and disease transmission. The authors failed to close the gap between the two imaging systems' capacities to identify specific autos. Transfer learning regularisation is incorporated into the goal function of a traditional convolutional neural network to achieve this. Experiments using the public data set of cars validated the effectiveness of the proposed method. Table 1 highlights the existing works carried out by several authors in the domain of smart cities as below:

Last research found several smart home surveillance gaps:

- The author found object detection accuracy limited in poor illumination or with obstructed objects.
- These methods cannot handle edge device video streams without sending sensitive data to the cloud or using privacy-preserving object detection methods.
- These studies failed to optimize smart home monitoring system energy consumption, notably with resource-intensive models like YOLO.
- These techniques were not require creating lightweight YOLO models or energy-efficient electronics.

| Table 2 | Dataset summary. | | | | |
|---------|------|--------|--------------|-------------|----------------|
| S. No. | Name | Images | Training set | Testing set | Validation set |
| 1 | Dataset 1 | 3,469 | 2,429 (70%) | 347 (10%) | 693 (20%) |
| 2 | Dataset 2 | 8,324 | 7,284 (88%) | 346 (4%) | 694 (8%) |

- They neglected to build defences to make YOLO more resistant to purposeful attacks.

# DATASET AND EXPERIMENT

## Dataset

In this work, the authors tested the model on two datasets from Roboflow. There are details of the images as below:

- Average image size: 0.55 mp
- Median image ratio: 960 × 576.

The details of the datasets are shown in Table 2 below:

Figure 1 shows the persons in the night from the camera installed in the home. These datasets consist of 2 different classes: infiltrator & person.

Setting up a YOLO model for smart home surveillance requires extensive data preparation. The quality and suitability of the input data for training and testing are checked here. Using the YOLO paradigm, here's how to preprocess data for smart home surveillance:

1. **Data collection**: Initially, the authors gathered a large sample of data from the smart home monitoring system, preferably in the form of still photographs and videos. To make the model as robust as possible, they should have scenarios with a wide range of lighting, item locations, and actions.
2. **Data labeling**: Labeling objects of interest in the dataset's pictures or video frames is "annotation". A class label and bounding box should be associated with every item. This is a crucial part of teaching the YOLO paradigm to identify new things.
3. **Data augmentation**: This step increases the variety of the dataset by using data augmentation methods. The model's applicability to real-world circumstances can be improved with the use of techniques such as random cropping, flipping, rotation, and variations in brightness.
4. **Data splitting**: This step creates a training set, a validation set, and a test set from the dataset. Training models typically make use of 70–80% of available data, with validation and testing making use of 10–20% each. Separating the data in this way aids in assessing the model's efficacy and avoiding overfitting.
5. **Data resizing**: This step reduces the size of every photo or video frame so that it fits the parameters of the YOLO model. To use a YOLO model, data must be resized to fit its required input dimensions, which are typically 416 × 416 or 608 × 608 pixels.
6. **Data normalization**: Images have their pixel values normalized such that they are all in the same range (like [0, 1] or [−1, 1]). Training the model using normalization helps it converge more quickly.
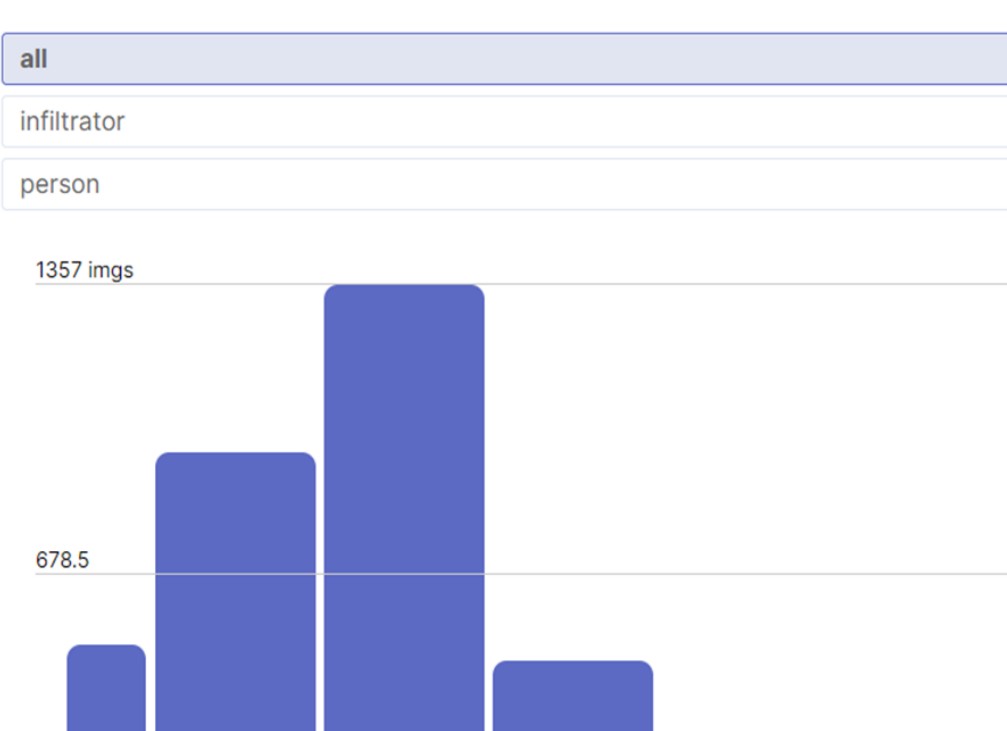

**Figure 1**  Different classes: infiltrator & person infiltrator.

7. **Data format conversion**: It is recommended to transform the labelled data into YOLO format, which generally comprises text files for each image, with each line containing the classified object's category, bounding box coordinates, and dimensions. Typical YOLO code looks like this: (class_id, x_center, y_center, width, height).

8. **Quality control**: This step checks the quality of the annotations and the photographs to make sure they are a good fit. Annotations that are inaccurate or inconsistent might have a detrimental effect on model training.

9. **Balancing class distribution**: To avoid having the model over- or under-predict for the dominant class, over- or under-sampling can be used if the dataset has an uneven class distribution.

10. **Data cleaning**: Finally, the authors maintain high data quality to necessitate the removal of any uncommon or useless data or pictures from the dataset.

In order to train a YOLO model for smart home monitoring, the data must first be adequately prepared.

## Methods

The field of deep neural networks, known as convolutional neural networks (CNNs), has seen impressive progress in recent years, notably with image classification tasks. By enabling computers to automatically and accurately identify objects in pictures, CNNs have transformed the area of computer vision (*Cai et al., 2023*; *Salman et al., 2017*; *Ramchandani, Fan & Mostafavi, 2020*; *Sooraj et al., 2020*; *Xu et al., 2020*; *Abdelali et al., 2021*; *Dash & Choudekar, 2021*; *Dhiyanesh et al., 2021*). CNNs' capacity to acquire hierarchical representations of information from the input pictures is the essential property that makes them particularly ideal for object categorization. Convolutional, pooling and fully linked layers are the building blocks of a standard CNN architecture (*García-González et al., 2021*). These procedures use filters to identify essential features for the object categorization process, such as edges, corners, and textures. The dissimilarity between the predicted and actual labels is often quantified using a loss function like a cross-entropy loss. Image classification, object recognition, and segmentation are just a few of the many object classification tasks where CNNs have excelled (*Harshith Kumar, Kaushik & Prajeesha, 2021*; *Xie et al., 2021*; *Zhou et al., 2021*; *Akhmetzhanov et al., 2022*). They find widespread usage in various contexts, including autonomous vehicles, surveillance, and medical imaging. The state-of-the-art YOLO model only employs a single neural network to forecast bounding boxes and class probabilities from entire photos. YOLO is ideally suited for real-time object identification applications because of its high efficiency and speed. YOLO predicts the bounding boxes and class probabilities for each cell in an image, unlike the sliding window technique used by most conventional object identification algorithms (*Alshamrani, 2022*). The model predicts the class and confidence score for each box within each cell, and the number of bounding boxes predicted by the model is constant across cells. A high confidence score suggests that an object is likely present in the box. In contrast, a high-class probability indicates that the object is expected to be a specific class member (*Chatterjee & Ahmed, 2022*; *Gupta, Pawade & Balakrishnan, 2022*; *Ibrahim et al., 2022*; *Li et al., 2022*).

The dynamic and complex nature of smart city surroundings makes it difficult to implement the YOLO (User Only Look Once) approach. The following factors may threaten the success of the YOLO model:

1. **Varying light conditions:** The day and nighttime lighting in smart cities are both different. The model's reliability in object detection can be affected by sudden shifts in illumination. User can get around this problem by using cameras with adaptive or dynamic exposure control and YOLO models optimized for low-light or high-contrast conditions.

2. **Object occlusions:** Anything in a public area has the potential to be obscured in whole or in part by anything else there. Correctly identifying objects might be complicated by occlusions. Implementing sophisticated object tracking algorithms in combination with YOLO can assist in retaining object identification and forecasting object movements even when they are momentarily concealed.

3. **Diverse human behaviors:** People in smart cities demonstrate a wide range of activities, from walking and cycling to participating in leisure pursuits. These actions may complicate object recognition. The capacity of a YOLO model to distinguish distinct behaviours may be enhanced by training it on datasets that encompass a wide range of human activities and postures.

4. **Complex backgrounds:** In smart cities, user'll find a wide variety of buildings, signage, and other things making up the scenery. The model may have trouble differentiating between real-world things and their surroundings. The YOLO model may be trained to exclude irrelevant items by using strategies like instance segmentation or backdrop removal.

5. **Traffic congestion:** Congestion on the roads is an everyday occurrence in cities. As a result, sceneries may get congested with numerous items that are too close together for the model to identify each one accurately. Accuracy in dense environments can be enhanced by employing object tracking and multi-object detection algorithms.

6. **Weather conditions:** Fog, rain, and snow can all diminish visibility and overall image quality. It is possible that the standard YOLO framework will not function well under these situations. Incorporating sensors like lidar or radar for supplemental object identification and developing YOLO models that are resilient to unfavourable weather conditions might improve performance.

7. **Privacy concerns:** The use of monitoring in smart cities gives rise to privacy worries. It might be challenging to strike a balance between security and privacy while collecting data in public settings. It is critical to solve privacy issues by using privacy-by-design concepts like real-time data anonymization and complying with data protection standards.

8. **Scalability:** Multiple cameras and sensors in a smart city can monitor a large region. It might be challenging to scale object detection models to a large number of sensors and real-time processing. The model's inference skills may be mounted to span large urban areas with the use of distributed computing and cloud-based technologies.

## Real-time processing

In order to quickly respond to crises or occurrences, real-time object detection is typically necessary in smart city applications. Maintaining minimal processing delays is not always easy to do. Real-time processing for object detection may be attained with the help of edge computing and high-performance hardware.

These complex problems in smart city settings can only be solved by a comprehensive strategy that incorporates state-of-the-art computer vision methods, sensor integration, and smart model design. The YOLO paradigm needs constant attention and refinement to function well in the ever-changing Context of modern cities. YOLO minimizes a loss function during training to make accurate predictions of the bounding boxes and class probabilities. Thousands of annotated photos of various items and backgrounds are used to train YOLO on massive datasets like common items in context (COCO) (*Madhuri et al., 2022*). With state-of-the-art performance on benchmarks like COCO, YOLO has proven to be a powerful tool for object recognition (*Madhuri et al., 2022*; *Pawar*

*& Attar, 2022*; *Sayed, Himeur & Bensaali, 2022*; *Yang et al., 2022*; *Ye et al., 2022*). YOLO has several uses, including self-driving cars, surveillance systems, and robots.

## Steps

The YOLO object detection model follows the following steps:

- *Input:* The YOLO model takes an image of arbitrary size and shape as input.
- *Preprocessing:* The input image is resized to a fixed size and normalized to a range of 0–1.
- *CNN:* The image is passed through a deep convolutional neural network, which extracts features from the image.
- *Grid:* The image is divided into a grid of cells, typically with a size of $13 \times 13$, $26 \times 26$ or $52 \times 52$.
- *Bounding boxes:* For each cell, the model predicts a fixed number of bounding boxes, normally 2, 5, or 10.
- *Class probabilities:* For each bounding box, the model predicts the class probabilities for all possible classes in the dataset.
- *Non-max suppression:* The model applies non-max suppression to remove redundant bounding boxes with low confidence scores and high overlap.
- *Output:* The final output of the YOLO model is a list of bounding boxes with associated class probabilities representing the objects detected in the input image.

The YOLO model is taught using a vast collection of annotated photographs featuring different foreground and background items and settings. The model learns to minimize a loss function to provide accurate predictions of the bounding boxes and class probabilities (*Park et al., 2022*). Typically, zlocalization loss, confidence loss, and classification loss are all used together as the loss function. Self-driving vehicles, surveillance systems, and robots are just a few of the many uses for the cutting-edge YOLO object detection model. Its speed and precision make it ideal for real-time object-detecting applications. There are several benefits to using this approach (*Reis & Stricker, 2023*).

The YOLO is renowned for its efficiency and speed. At 45 frames per second, it can do real-time image processing on a GPU. Second, YOLO has a high degree of accuracy when it comes to object identification tasks, particularly for medium and tiny items. COCO is only one of several benchmarks that have proved how much better it is than other top-tier object identification models. Finally, a single neural network is utilized to forecast bounding boxes and class probabilities from entire pictures in YOLO (*Eslick et al., 2023*; *Himeur et al., 2023*; *Madhu et al., 2023*; *Nurnoby & Helmy, 2023*). Thus, there is no need for a region proposal phase. As compared to previous models for object detection, this one doesn't require as many steps. Finally, YOLO may be readily modified to incorporate object tracking by linking together items identified in different frames. Fifth, YOLO may be set up to recognize specific types of objects (humans, cars, animals, *etc.*).

This approach has drawbacks, such as its inability to identify minute details in a picture accurately. This mishap occurred because the model predicts bounding boxes using a predetermined grid of cells, which may not be sufficient for adequately localizing highly

tiny objects. Objects substantially obscured by others in the scene may likewise be complex for it to identify (*Prazeres et al., 2023*). The model makes predictions about bounding boxes for each grid cell in isolation from the rest of the picture. The model's ability to generalize to the novel, unknown data may be compromised if the training dataset favours some classes of objects or situations over others. False positives, or the identification of items that are not actually there, can occur while using YOLO. This can happen if the model is not trained with data that adequately represents real-world conditions. When applied to huge photos or when executed in real-time, YOLO places heavy demands on the computing resources available. This might restrict its usefulness in settings with scarce resources.zzz. Hence, YOLO is a highly effective and popular object detection model. Still, it has some limitations that need to be considered when using it in real-world applications. There are several ways to optimize this model, including:

- *Adjusting the hyperparameters*: Several hyperparameters in the YOLO model can be tweaked to enhance its functionality. The training epochs, batch size, and learning rate are all.
- *Data augmentation*: Image transformations (including flipping, rotating, and scaling) are examples of data augmentation techniques that may increase the diversity and amount of the training data, hence bolstering the model's precision and stability.
- *Transfer learning*: For object recognition applications, the YOLO model may be trained on a large dataset like ImageNet and then fine-tuned on a smaller dataset using transfer learning. This approach has the potential to significantly shorten training times while also increasing the model's precision.
- *Pruning*: The YOLO model may be pruned to make it smaller by eliminating excessive weights and neurons. This approach might make the model quicker and use less memory.
- *Quantization*: With Quantization, the YOLO model's speed and memory footprint may be improved without considerably sacrificing accuracy by decreasing the precision of the weights and activations.
- *Hardware acceleration*: The YOLO model can be optimized for specific hardware architectures, such as GPUs or FPGAs, to improve performance and reduce power consumption.

Optimizing the YOLO model requires technical expertise and experimentation to find the optimal combination of hyperparameters, data augmentation, and optimization techniques.

## Proposed methodology

The YOLO model can benefit greatly from including transfer learning in order to excel at specific object identification tasks. The similarities between the source and destination domains are typically taken for granted in transfer learning. Domain shift can occur when there is a large gap between the source dataset used for pre-training and the target domain in terms of object categories, lighting conditions, or object positions. Without resolving this issue, the YOLO paradigm may not transfer effectively to the new domain. It can be time-consuming and expensive to annotate a dataset for transfer learning. For tuning,

having access to a labelled collection of data from the target domain is crucial. It might be challenging to acquire or generate a representative dataset for the smart home monitoring assignment. To predict both bounding boxes and class probabilities concurrently, YOLO suggests employing an end-to-end neural network. In contrast to other algorithms, YOLO uses a single, fully connected layer to make predictions. The YOLO technique requires an image to be partitioned into N grids, each of which is a uniformly sized S × S square.

### Algorithm 1 YOLO Algorithm

Step 1.   *Train the model using floating-point precision:* The YOLO model is first trained using the standard floating-point precision, which provides high accuracy but requires a lot of memory and computational resources.

Step 2.   *Analyze the weight and activation distributions:* Once the model is trained, weights and activations are analyzed to determine the appropriate quantization levels and scaling factors.

Step 3.   *Apply quantization to the weights:* This is done using the following equation: $q = \text{round}(w/s)$, where q is the quantized value, w is the continuous-valued weight, s is the scaling factor, and round() is the rounding function.

Step 4.   *Apply quantization to the activations:* The activations in the YOLO model are also quantized using a similar process. Each activation value is represented as an 8-bit integer and scaled to fit within the range of the quantization levels.

Step 5.   *Re-evaluate the performance of the quantized model:* Once the model has been quantized, it is re-evaluated on a validation set to determine the impact of Quantization on the model's accuracy.

Step 6.   *Fine-tune the quantized model:* If the performance of the quantized model is not satisfactory, it can be fine-tuned by retraining the model using the quantized weights and activations.

Step 7.   *Deploy the quantized model:* Finally, the YOLO model can be deployed on resource-constrained devices, providing high-quality object detection with reduced memory requirements and improved inference speed.

Integrating transfer learning with the YOLO model for smart city surveillance has an excellent future scope. As the development of smart cities continues to grow, the demand for accurate and efficient object detection algorithms for surveillance applications will only increase. Transfer learning can help address some of the challenges associated with training object detection models on limited data. One future direction for integrating Transfer learning with the YOLO model for smart city surveillance is the development of more advanced transfer learning techniques. For example, recent research has explored domain adaptation and meta-learning techniques to improve the transferability of pre-trained models to new domains. These techniques could be applied to the YOLO model for smart city surveillance to improve its accuracy and adaptability to new environments. Figure 2 shows one possible implementation of YOLO for identifying many items at once.

Quantization can be applied to the YOLO object detection model, similar to other deep neural networks. Let's consider the weight tensor W in the YOLO model with shape (N, C, H, W), where N is the batch size, C is the number of channels, H is the height, and W is the

width. Quantization involves representing each weight value w in W as an 8-bit integer q by rounding it to the nearest integer and scaling it to fit within the range of the quantization levels. When determining the quantization levels, the range of the weights is divided into 256 equal intervals, with 0 representing the minimum value and 255 representing the maximum value. The weight range is divided by the quantization level range to get the scaling factor. The process of quantization can be mathematically modelled using the following equation:

$$q = round\left(\frac{w}{s}\right) \tag{1}$$

where q is the quantized value, w is the continuous-valued weight, s is the scaling factor, and round() is the rounding function. The resulting quantized weights are stored in an 8-bit integer format, reducing the model's memory requirements. In addition to quantizing weights, activations can be quantized using a similar process. Quantizing activations involves representing each activation value as an 8-bit integer by rounding it to the nearest integer and scaling it to fit within the range of the quantization levels. The scaling factor is determined by analyzing the distribution of the activations during the training phase. Overall, the process of quantization can be applied to the YOLO model, similar to other deep neural networks. By reducing the precision of the weights and activations, quantization can significantly reduce the memory requirements and improve the inference speed of the YOLO model, making it more suitable for deployment on resource-constrained devices.

Pruning entails locating and eliminating insignificant model parameters, especially those with negligibly tiny weight magnitudes. This decreases the model's storage and processing needs. The YOLO model's neural network design can benefit from pruning, which gets rid of unused nodes and weights. It is more effective for real-time object identification since it can drastically decrease model size and increase inference speed. A smaller "student" model is trained to mimic the actions of a more prominent "teacher" model; this is known as "knowledge distillation." The teacher's predictions serve as a learning tool for the student model, allowing it to improve its generalization and efficiency. In the YOLO framework, a more condensed model may be trained to simulate the performance of a more comprehensive model in terms of object detection. The condensed model can then be utilized for immediate inference while making use of the instructor's expertise. Another future direction is the development of more specialized pre-trained models for smart city surveillance. For example, pre-trained models could be developed to detect objects of interest in different smart city environments, such as traffic intersections or pedestrian walkways. These specialized models could be fine-tuned on smaller datasets of images from these environments to improve their accuracy and efficiency. Integrating Transfer learning with the YOLO model for smart city surveillance has a bright future, with many opportunities for further research and development. As the demand for accurate and efficient object detection algorithms continues to grow, transfer learning techniques could help to drive progress in this area and enable the development of more effective surveillance systems for smart cities. When integrating transfer learning with YOLO, the

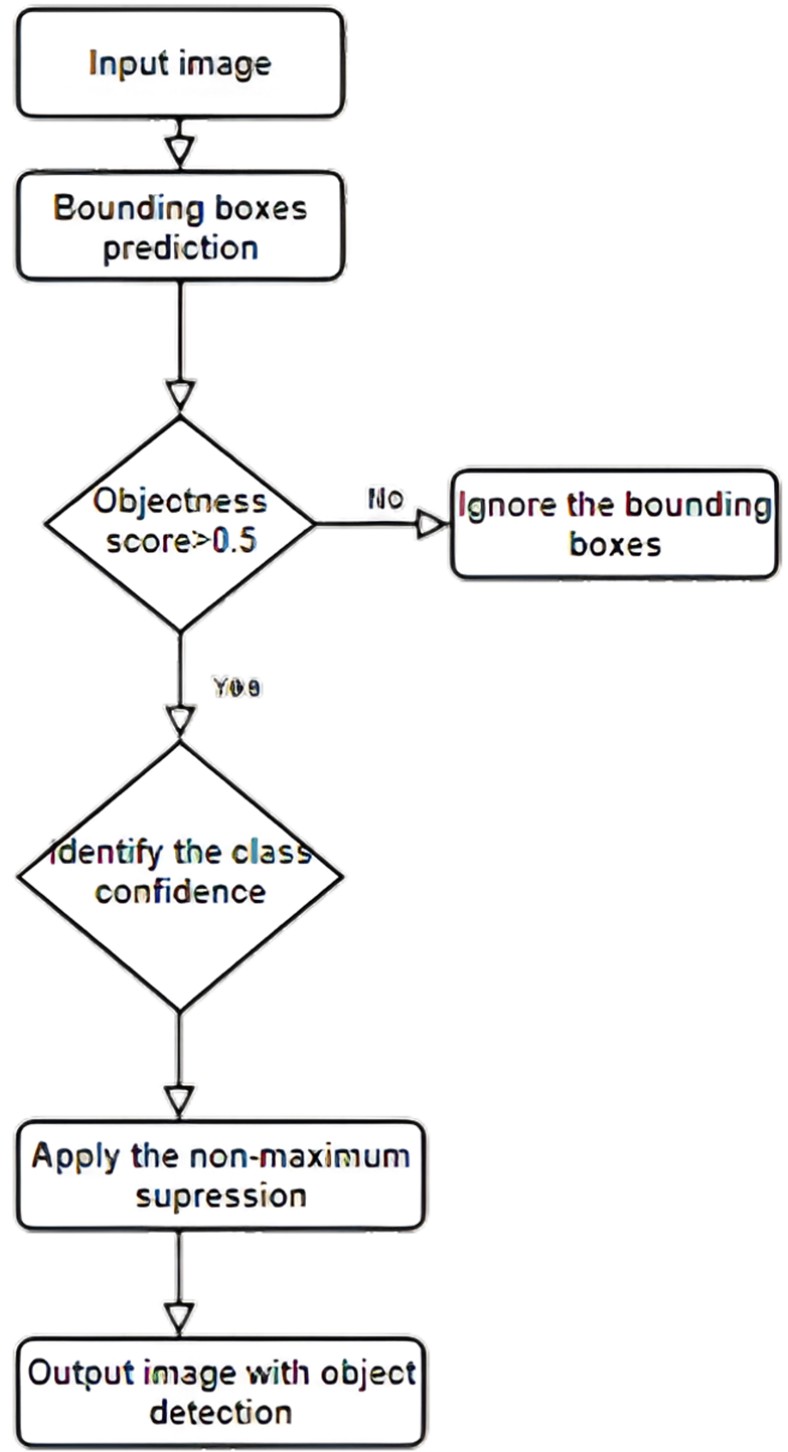

**Figure 2** Multiple object detection mechanism using YOLO.

YOLO architecture may need to be modified to suit the new object detection task. Here are some common modifications that can be made:

### Algorithm 2 proposed YOLO algorithm

Step 1.      *Change the number of classes:* The number of classes in the pre-trained YOLO model may differ from those in the new object detection task. Therefore, the final layer of the network must be modified to output the correct number of classes.

Step 2.      *Change the input size:* The input size of the pre-trained YOLO model may not match the input size of the new dataset. Therefore, the input size of the YOLO model must be modified to match the input size of the new dataset.

Step 3.      *Add or remove layers:* The YOLO architecture may be modified by adding or removing layers depending on the complexity of the new object detection task. For example, if the new dataset contains objects of smaller size, it may be necessary to add more convolutional layers to capture more fine-grained features.

Step 4.      *Adjust hyperparameters:* The hyperparameters of the YOLO model, such as the learning rate, batch size, and the number of epochs, may need to be adjusted to achieve better performance on the new object detection task.

Modifying the YOLO architecture is an essential step in integrating Transfer learning with YOLO as it ensures that the model is optimized for the new object detection task.

## RESULTS AND ANALYSIS

### Experiment setting

In this research work, a laptop with a 2.40 GHz Intel Core i3-4000M CPU and 4 GB of RAM was used with the Tesla K80 GPU. The Keras and Tensor-Flow Python libraries, both open-source DL software packages, were employed in this study. The authors also used the statistics package Scikit-learn to assess the performance metrics.

### Experimental results

Figure 3 shows training cls_loss. The classification loss measures the error in the predicted class probabilities for each bounding box. It is calculated using the cross-entropy loss function, which penalizes the model for assigning high probabilities to incorrect classes and low probabilities to correct classes.

Figure 4 shows training box_loss. In YOLO, the object loss is calculated for each grid cell that contains an object. Classification loss and localization loss are combined to form the loss function. Classification loss quantifies how much each object's anticipated class probabilities deviate from its actual ground-truth class label. The localization loss is the deviation between the expected and ground-truth bounding box positions.

The object loss term is essential if user want more reliable object recognition using the YOLO model. During training, the model adjusts its weights to minimize this loss and produce more accurate predictions of object presence, location, and category in the input image. The choice of hyperparameters, such as $\lambda_{obj}$, can significantly impact the model's

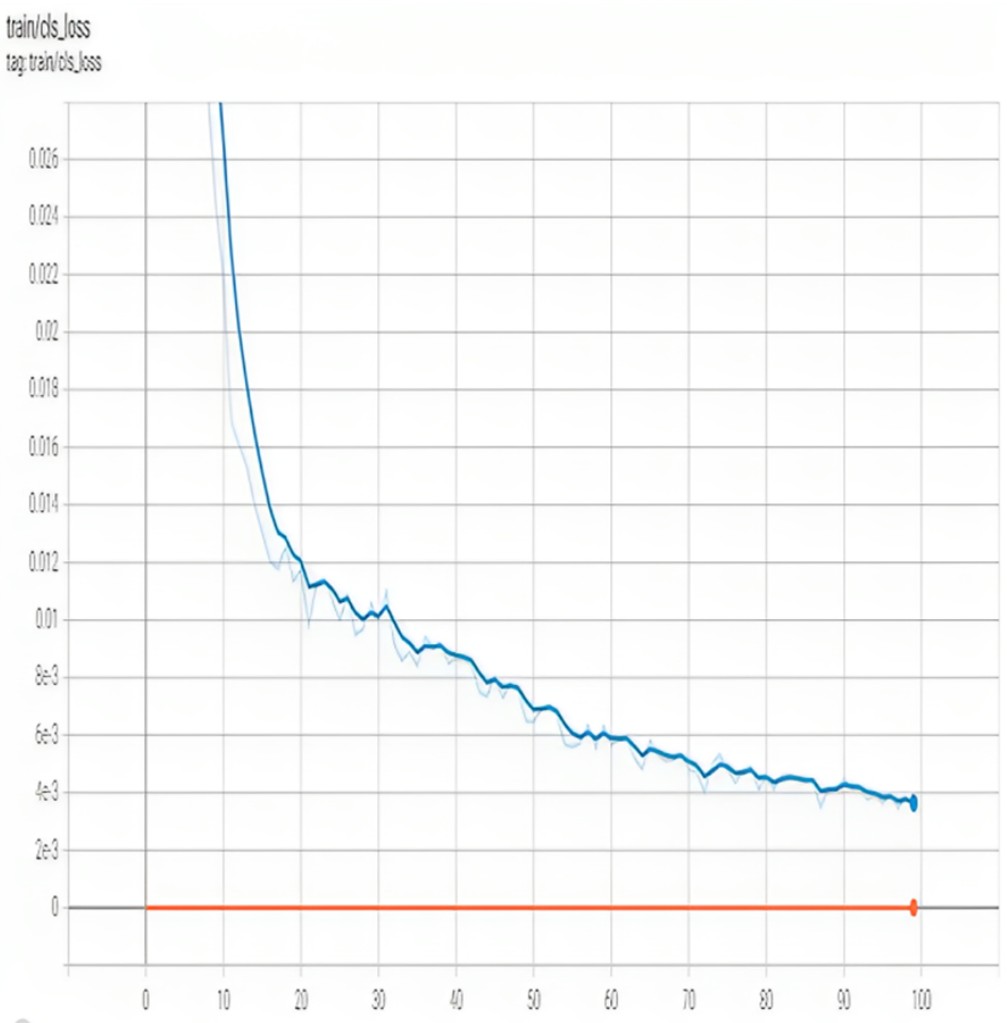

**Figure 3  Training Cls loss of of proposed model.**

performance, and careful tuning is often necessary to achieve optimal results. The accuracy of a model may suffer if it is optimized for efficiency (through quantization, for instance). The authors may have to compromise some detection precision to maximize speed and efficiency. Quantization, the process of decreasing the number of bits in a bit of data, can aid in the deployment of models by lowering their associated memory and processing demands. However, the model's ability to pick up nuanced information can suffer due to this decrease in precision. The low bit-depth of quantized data might cause some blurring or distortion of form in some circumstances. It might be challenging to apply transfer learning with a limited smart home monitoring dataset. The model might be overfitting the data from the input domain. There may be a mismatch between the source and target domains; however, using a larger dataset from the source domain can assist.

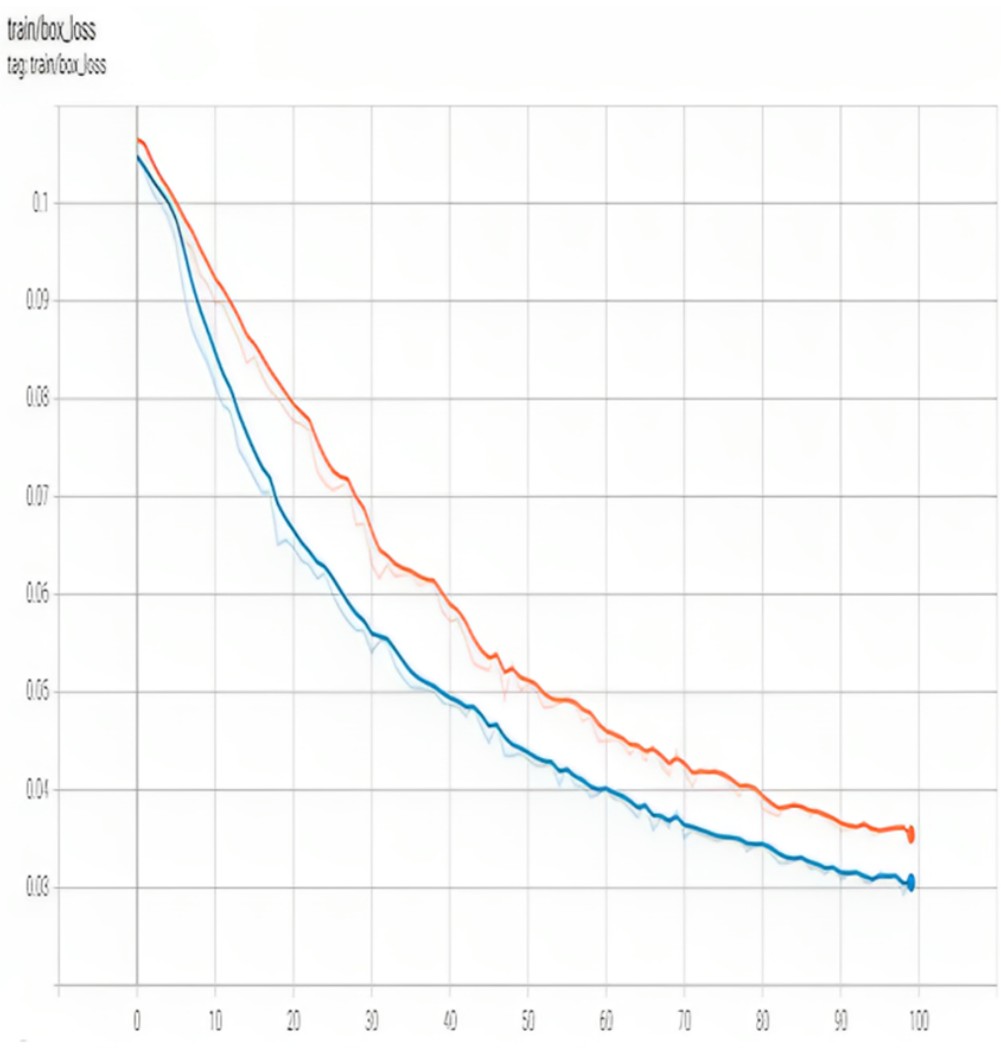

**Figure 4** **Training box loss of proposed model.**

## Classification measure

The model's performance may be dissected and comprehended using the following variables.

$$Accuracy = \frac{TP + TN}{TP + TN + FP + FP} = \frac{Correct\ predictions}{Total\ predictions} \tag{2}$$

$$Precision = \frac{TP}{TP + FP} = \frac{Predictions\ actually\ positive}{Total\ predicted\ positive} \tag{3}$$

$$Recall = \frac{TP}{TP + TN} = \frac{Predictions\ actually\ positive}{Total\ actual\ positive} \tag{4}$$

**Table 3  Classification measure.**

| Optimizer | Images | Precision | Recall | mAP@.5 | mAP@.5:.95 |
|---|---|---|---|---|---|
| SGD Optimizer | 8,324 | 0.857 | 0.816 | 0.055 | 0.019 |
| Bayesian Optimizer | 8,324 | 0.935 | 0.830 | 0.704 | 0.349 |
| Quantisation | 8,324 | 0.988 | 0.908 | 0.819 | 0.569 |

Classification measures for SGD, Bayesian, and Quantization optimizers using the Yolo model to categorize meteorological conditions are shown in Table 3.

Achieving high precision in a YOLO object detection model requires careful consideration of various factors such as the number of training epochs, learning rate, training data, data augmentation, fine-tuning, confidence threshold, and object scale. By implementing the techniques mentioned earlier, user can improve the precision of the YOLO model, making it more accurate in detecting objects in images. It is worth noting that achieving high accuracy in object detection models is a challenging task that requires both theoretical knowledge and practical experience. Therefore, user must keep experimenting with different techniques and adjusting the model parameters until user find the optimal configuration that works best for the specific use case. Figure 5 shows precision.

Recall is an essential metric for evaluating the performance of object detection models, including YOLO models. It measures the model's ability to correctly identify all instances of a given class in the image. Figure 6 shows recall.

By implementing these techniques, the researchers can improve the recall of the YOLO model, making it more effective in detecting all instances of a given class in an image. It's important to note that achieving high recall is a challenging task that requires experimentation with different techniques and adjustments to model parameters. Therefore, it's essential to carefully evaluate the model's performance and adjust as needed to achieve the desired level of recall for the specific use case.

Standard metrics for object detection accuracy often include average precision (AP). One metric that may be used to assess this is the region lying below the accuracy-recall curve. It is calculated using the formula:

$$AP = \int_0^1 p(r)\, dr. \tag{5}$$

Figure 7 illustrates mAP_0.95. YOLO mAP@.5 is a popular metric used to evaluate the accuracy of object detection models, including YOLO. It represents mean average precision at a 0.5 intersection-over-union (IoU) threshold. The mAP@.5 measures the percentage of correctly detected objects when the predicted bounding box overlaps with the ground truth determining box by at least 50%. A higher mAP@.5 indicates better accuracy in object detection. YOLOv3, one of the latest versions of the YOLO model, has achieved a mAP@.5 score of 57.9% on the COCO dataset, a widely used benchmark for object detection models.

Figure 8 illustrates mAP@.5:.95. The mAP@.5:.95 is another popular metric used to evaluate the accuracy of object detection models, including YOLO. It is the mean average precision overall object categories for detection bounding boxes, with IoU ranging from 0.5 to 0.95. The mAP@.5:.95 measures the percentage of correctly detected objects

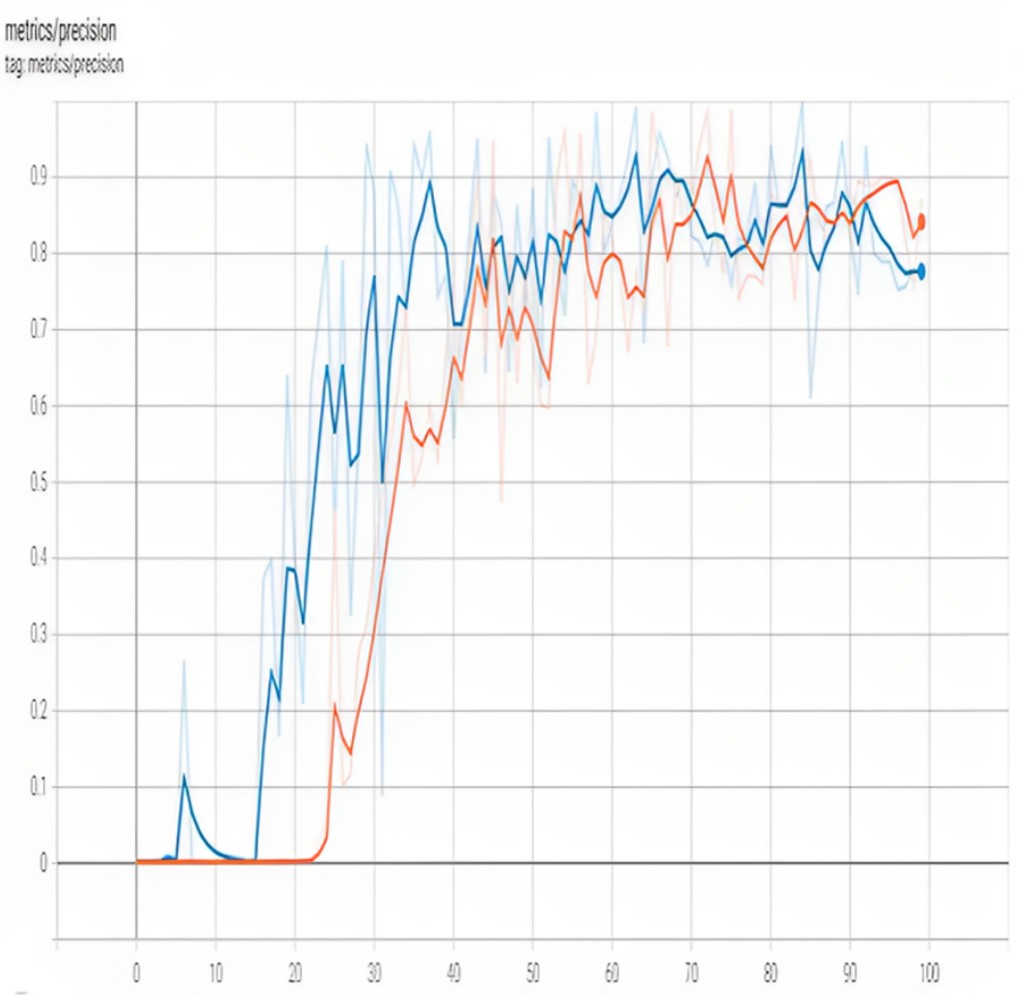

**Figure 5** Precision of proposed model.

across different levels of overlap between the predicted bounding box and the ground truth bounding box. A higher mAP@.5:.95 indicates better accuracy in object detection. YOLOv4, one of the latest versions of the YOLO model, has achieved a mAP@.5:.95 score of 43.5% on the COCO dataset, which is an improvement over its predecessor YOLOv3 that achieved a score of 39.0%.

## Training loss

The accuracy with which an item is classified using the YOLO model is shown by a metric known as the classification loss. Each recognized item in an image contributes to the overall classification loss, which is the discrepancy between the anticipated and ground-truth class probabilities. When determining classification loss, the YOLO model employs the cross-entropy loss function. This metric measures the amount of misfitting between the training data and the deep learning model. Figure 9 shows train obj_loss.

0.11
0.075
0.2
0.02

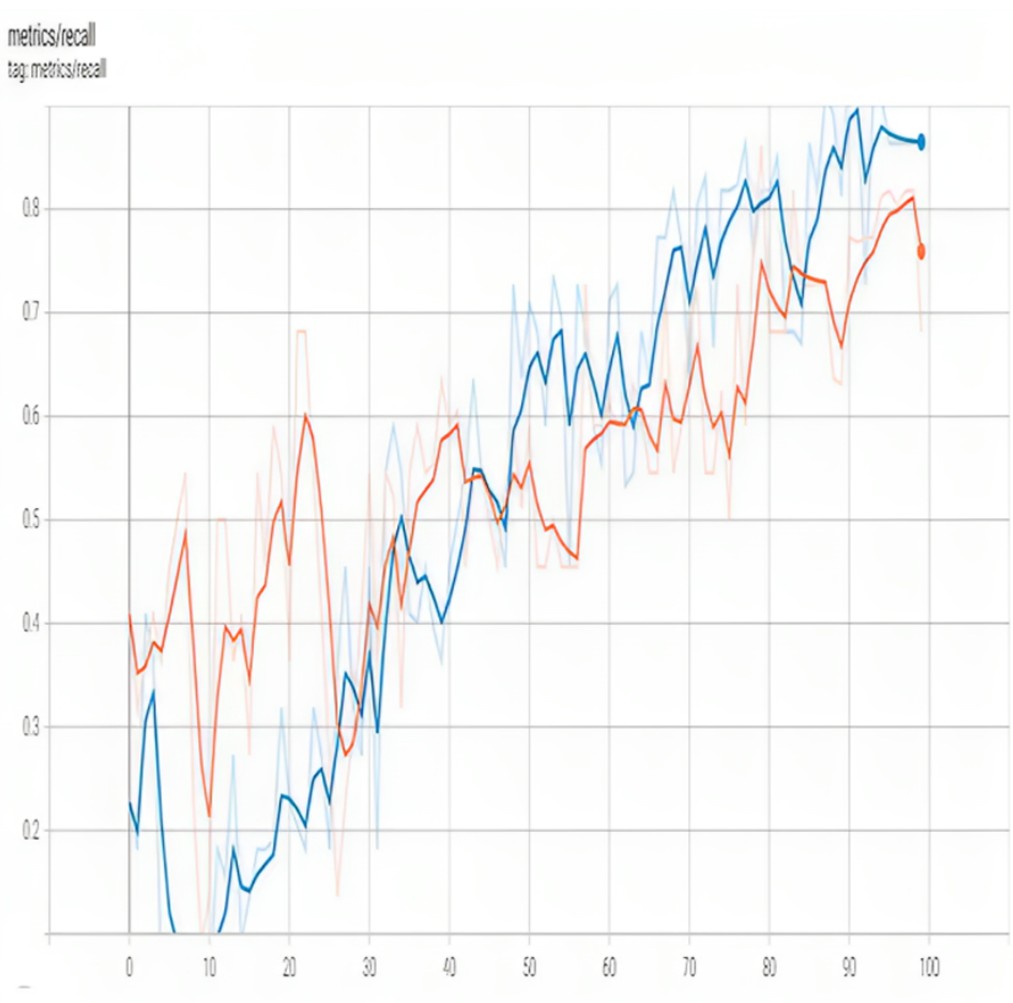

**Figure 6  Recall of proposed model.**

Figure 10 shows the confusion matrix for the current problem of smart city surveillance. Accuracy performance metrics for several approaches, including the suggested method, are shown in Table 4. The proposed model was found to improve upon the accuracy of previous models.

The proposed model is immune to environmental noise and adverse weather conditions, allowing it to distinguish between humans, animals, and objects and immediately notify security personnel of potential threats. Table 5 illustrates the implementation of different YOLO model variants based on accuracy performance metrics. Table 5 demonstrates the impact of the accuracy gained by the proposed model. The proposed model was found to improve upon the accuracy of previous variants of the YOLO model which achieved the answer of the research question RQ1 and proved its hypothesis.

This function is essential for enhancing security procedures by improving detection quality, decreasing false positives, expanding the field of vision, and so on. With deep

Dalal et al. (2024), *PeerJ Comput. Sci.*, DOI 10.7717/peerj-cs.1939      21/34

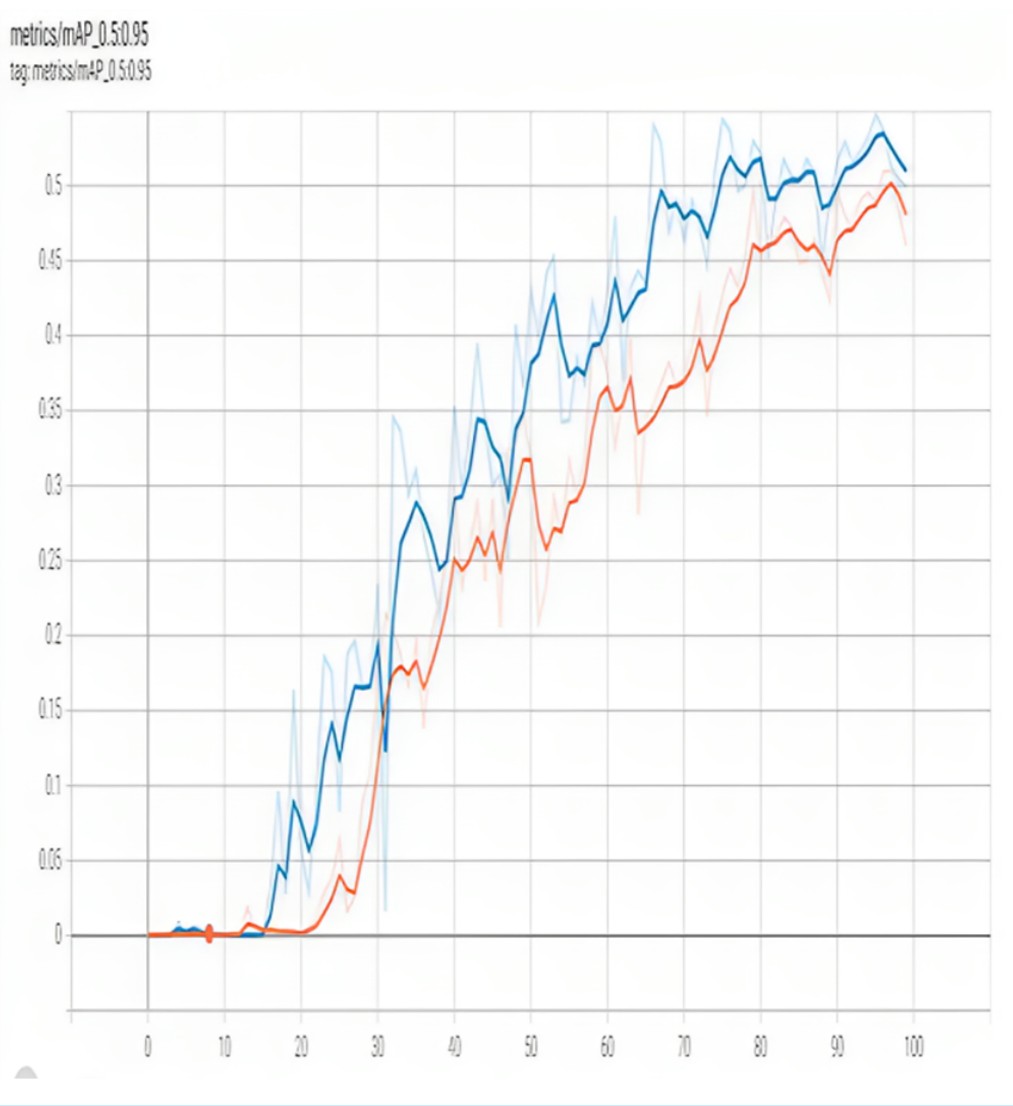

**Figure 7** MAP 0.95 of proposed model.

learning, security professionals may have immediate two-way conversations with intruders and suspects while reducing false positives.

Table 6 demonstrates the results of various state-of-the-art models and the proposed method regarding precision and recall for every class. It has been observed that the proposed model gained higher accuracy than existing models which achieved the answer of the research question RQ2 and proved its hypothesis.

The implications of integrating Transfer learning with the YOLO model for smart city surveillance can be significant. Here are some potential implications:

- *Improved accuracy:* Transfer learning can help improve object detection and classification accuracy in smart city surveillance applications. By leveraging the knowledge gained from pre-trained models on large and diverse datasets, transfer

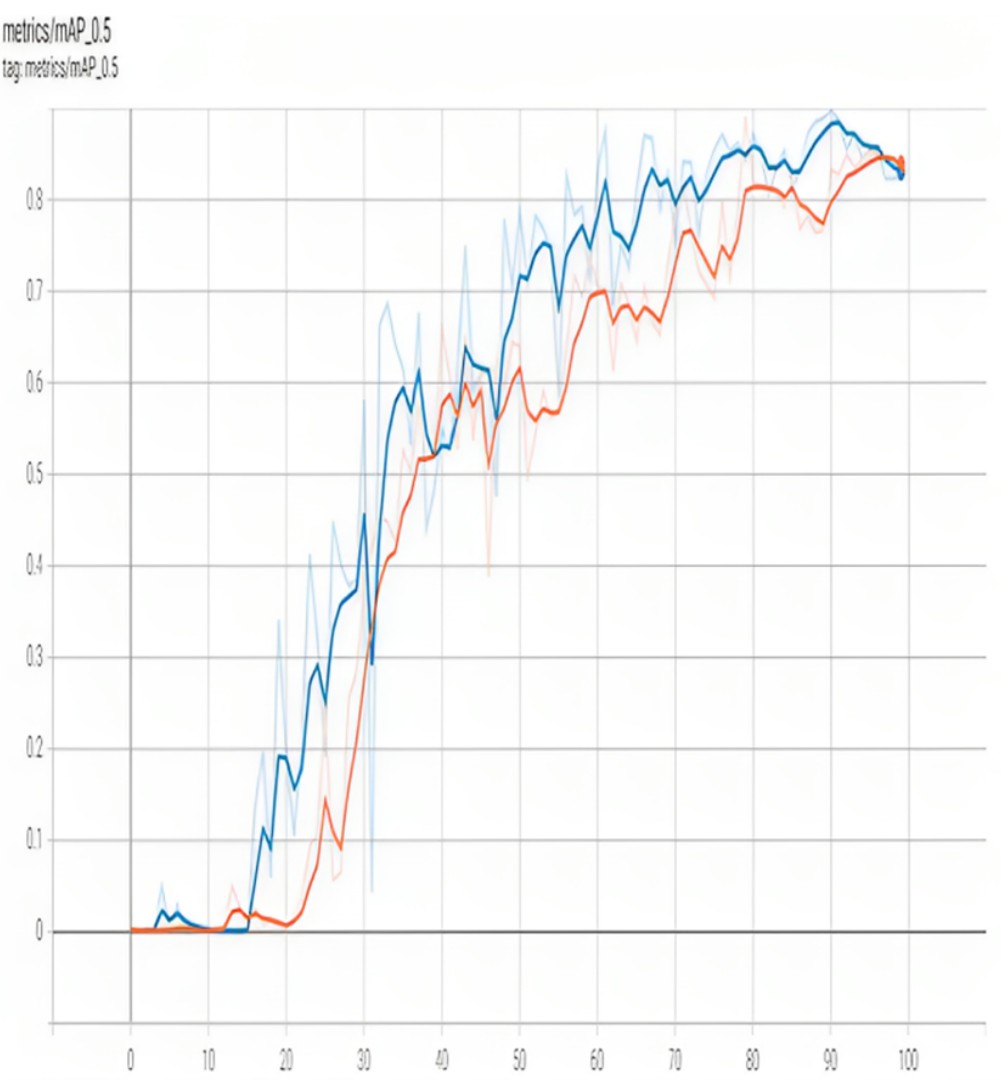

**Figure 8** MAP 0.5 of proposed model.

learning can help to address the challenge of limited data in smart city surveillance datasets.

- *Reduced training time:* By starting with a pre-trained model, the transfer learning approach can reduce the time and computational resources required for training the YOLO model on the smart city surveillance dataset.
- *Customization*: Transfer learning allows for customizing pre-trained models better to suit the specific requirements of smart city surveillance applications. For example, the model can be fine-tuned to detect specific objects or classes relevant to the surveillance task.
- *Transferability:* Transfer learning can enable the transfer of knowledge gained from one smart city surveillance task to another, potentially reducing the need for large amounts of annotated data for each task.

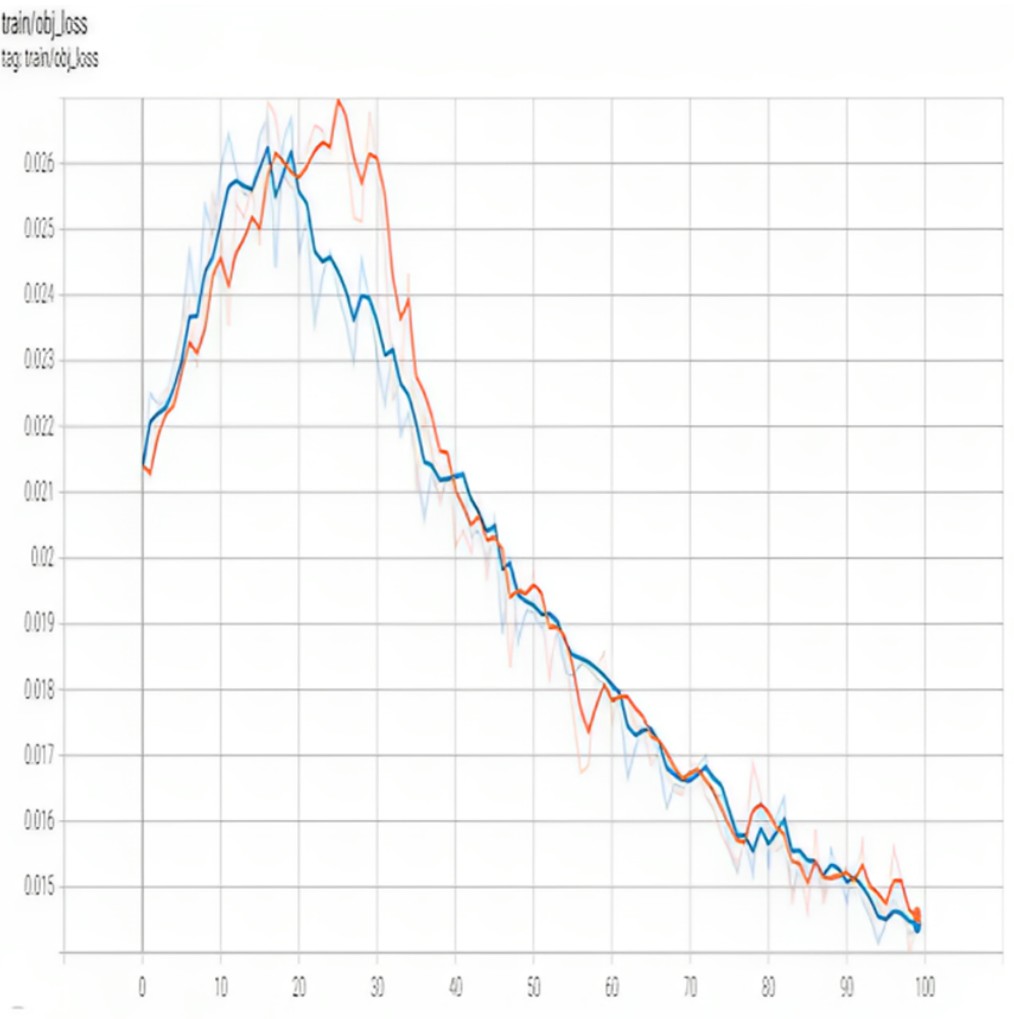

**Figure 9 Train obj_loss of proposed model.**

- *Real-time performance:* The YOLO model is known for its ability to provide real-time object detection. Integrating transfer learning can help optimize the model for real-time performance in smart city surveillance applications.

Integrating transfer learning with the YOLO model for smart city surveillance can significantly improve the accuracy, efficiency, and customization of object detection and classification in this domain. Applying a YOLO model with transfer learning and quantization to smart home monitoring may have far-reaching consequences and several practical uses, including the following.

1. **Enhanced security and intrusion detection:** Transfer learning and quantization can improve object identification accuracy, boosting smart home security. The model can accurately identify intruders, facilitating prompt notifications and responses, such as sending out alerts to homes or setting off alarms which aligned with RQ1.

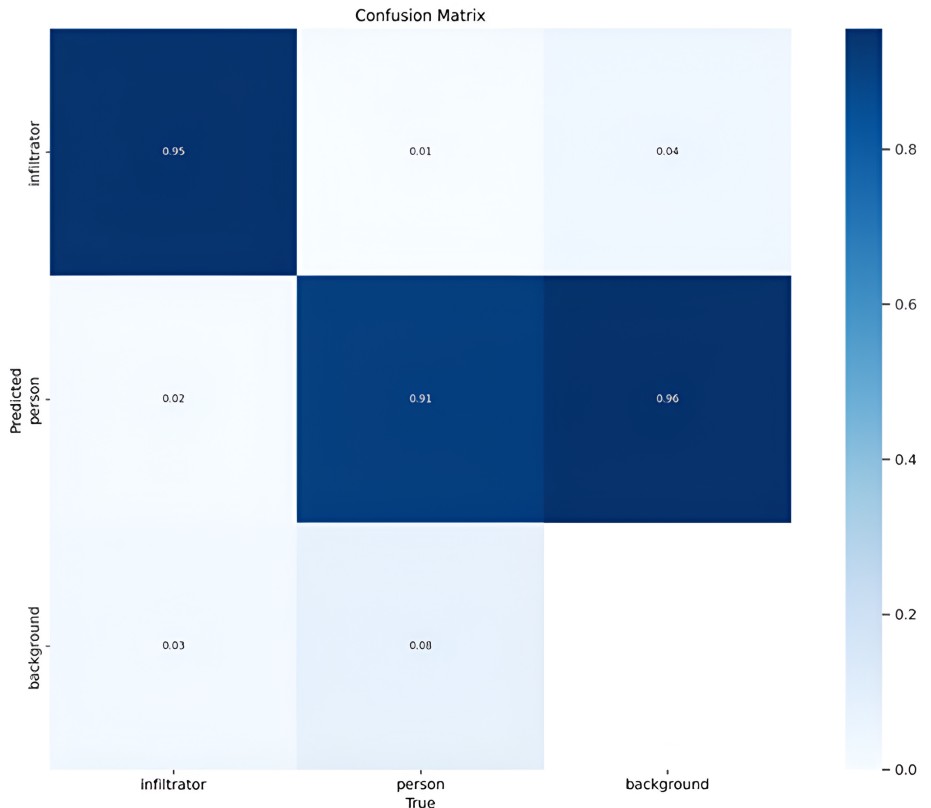

**Figure 10** Confusion matrix of proposed model.

**Table 4 Result comparison.**

| S. No. | Article | Methods | Result |
|---|---|---|---|
| 1 | *Wang et al. (2018)* | CNN and transfer learning | 55.13% Accuracy |
| 2 | *Kaşkavalci & Gören (2019)* | Deep Learning | 93.56% Accuracy |
| 3 | *Abdelali et al. (2021)* | YOLO detection approach | 92.50% Accuracy |
| 4 | *Ahmed et al. (2021)* | YOLO model | 94% Accuracy |
| 5 | *Wu et al. (2021)* | ResNet-50 model | 94% Accuracy |
| 6 | *Sivachandiran, Jagan Mohan & Mohammed Nazer (2022)* | EfficientDetmodel | 92.95 Precision |
| 7 | *Kulurkar et al. (2023)* | LSTM model | 95.87% Accuracy |
| 6 | Proposed model | Yolo7 model with transfer learning | 98.87% Accuracy |

2. **Intruder classification and alerts:** Intruders can be identified as known (family, visitors) or unknown (strangers) using the model's object recognition skills, which go beyond simple detection. It can use this information to issue specific warnings or notifications which aligned with RQ2.

3. **Fire and hazard detection:** It is possible to teach the YOLO model to detect the presence of fire, smoke, and other dangers. This may be very helpful for early detection and prompt actions like setting off fire alarms or calling for help.

**Table 5  Performance of different YOLO variants on same dataset.**

| Yolo variants | Images | Precision | Recall | mAP@.5 | mAP@.5:.95 |
|---|---|---|---|---|---|
| YOLOv2 model | 8,324 | 0.751 | 0.715 | 0.057 | 0.011 |
| YOLOv3 model | 8,324 | 0.851 | 0.801 | 0.410 | 0.28 |
| YOLOv4 model | 8,324 | 0.904 | 0.701 | 0.651 | 0.318 |
| YOLOv5 model | 8,324 | 0.935 | 0.830 | 0.704 | 0.349 |
| Proposed Yolo model | 8,324 | 0.988 | 0.908 | 0.819 | 0.569 |

**Table 6  Precision and recall for every class.**

| Parameters | EfficientDet model | | MobileNet model | | Proposed model | |
|---|---|---|---|---|---|---|
| | Person | Infiltrator | Person | Infiltrator | Person | Infiltrator |
| Precision | 91.25% | 89.64% | 93.26% | 94.25% | 98.17% | 98.91% |
| Recall | 88.91% | 86.25% | 92.65% | 93.41% | 97.65% | 97.35% |
| Accuracy | 90.34% | 88.26% | 91.47% | 92.57% | 98.15% | 98.36% |
| F1-score | 89.67% | 89.31% | 92.57% | 92.78% | 92.16% | 93.26% |
| Specificity | 88.26% | 87.46% | 90.87% | 93.71 | 95.18% | 97.68% |
| AUC-ROC | 89.26% | 88.56% | 89.78% | 91.24% | 94.36% | 97.86% |

4. **Child and pet watching:** Pet and kid surveillance is possible with a smart home system. The model's ability to keep tabs on kids and animals gives parents much-needed peace of mind.

5. **Automated homes and situational awareness:** The model's object identification features are compatible with existing home automation infrastructure. When people enter a room, the system can detect their presence and adjust the environment.

6. **Energy efficiency:** Object detection can help smart homes save money on energy bills. If the model knows which rooms are inhabited, it can turn down the thermostat or turn off the lights to save money.

7. **Tracking of shipments and delivery:** The YOLO model may be set up to monitor the arrival and departure of delivery persons and to identify when packages have been delivered. This is a helpful and convenient addition for homes since it may aid with package security.

8. **Health and well-being monitoring:** The program may be modified to track the well-being and routines of the elderly and other at-risk people. Improve healthcare and well-being services by sending out alarms in the event of falls or odd inactivity.

9. **Privacy and data protection:** Smart home monitoring data should allay fears about personal data leakage. To reduce privacy risks and stay in compliance with data protection requirements, it is recommended to implement tools that automatically obscure or obfuscate faces and sensitive information.

10. **Real-time alerts and remote control:** Homeowners can receive real-time notifications from the model on their smartphones or other devices. This way, user can monitor and adjust the smart home's settings without leaving the office.

11. **Adaptive lighting and security features:** Adaptive lighting, which ensures well-lit routes and alerts people of their presence in particular places, may be provided by integrating object detection with lighting and security systems.

12. **Custom object recognition:** The YOLO algorithm may be tweaked to identify everything from everyday things to high-value possessions and even abnormal occurrences like water leaks.

Surveillance in a smart home using a YOLO model with transfer learning and Quantization has far-reaching ramifications beyond safety, including automation, convenience, and well-being. As these technologies grow increasingly commonplace in private homes, careful deployment and respect for residents' privacy are essential. Overall, precise object detection coupled with efficient computational resources can improve the quality of life in smart homes by making them safer and more efficient. To ensure ethical technology use, YOLO with transfer learning and quantization for smart home surveillance must solve various ethical issues. Important ethical considerations are:

- The surveillance system must be approved by all residents before being put in their homes. Establishing standards for monitoring system data consumption, dissemination, and retention and identifying its owner are essential.
- To protect sensitive data from hackers and others, data must be stored with strong encryption techniques that only authorized personnel can interpret.
- To avoid biased findings, especially regarding gender, race, and other sensitive factors, it must repair any biases in the YOLO model's training data and regularly check for and eradicate monitoring system biases.
- Users should comprehend and follow the YOLO model's decision-making process. Publicly sharing the model's architecture, training data, and evaluation metrics promotes accountability and transparency, helping users trust and understand the system.

By considering these moral concerns, smart home surveillance system developers will improve people's capacity to use technology responsibly and ethically while maintaining their privacy.

## CONCLUSIONS

As part of this effort, the authors developed an optimized YOLO model with transfer learning using quantization to identify out-of-the-ordinary occurrences. Integrating transfer learning with the YOLO model for smart city surveillance has several advantages. First, transfer learning allows the YOLO model to be trained on a smaller dataset, which reduces the need for a large amount of labelled data. This benefit is significant for smart city surveillance applications, where collecting and labelling a large amount of data can be difficult and expensive. Second, transfer learning allows the YOLO model to be adapted to specific smart city surveillance use cases. For example, suppose the smart city surveillance application focuses on detecting particular objects, such as vehicles or pedestrians. In that case, the YOLO model can be fine-tuned on a smaller dataset of images containing those specific objects. Third, transfer learning can improve the accuracy of the YOLO model. By starting with a pre-trained model and fine-tuning it on a smaller dataset, the model

can learn to recognize features and patterns specific to the smart city surveillance use case. However, there are also some limitations to integrating transfer learning with the YOLO model for smart city surveillance. One limitation is that transfer learning requires a pre-trained model already trained on a large dataset. If such a pre-trained model is not available, implementing transfer learning may not be easy. Another limitation is that fine-tuning the YOLO model on a smaller dataset can lead to overfitting. The model becomes too specialized to the training data and does not generalize well to new data. To avoid overfitting, it is essential to carefully select the pre-trained model and fine-tune it on a representative subset of the smart city surveillance dataset.

In conclusion, integrating transfer learning with the YOLO model for smart city surveillance can potentially improve the accuracy and efficiency of object detection in these applications. However, it is essential to carefully consider the advantages and limitations of transfer learning and design the fine-tuning process to avoid overfitting. Integrating transfer learning with the YOLO model for smart city surveillance has an excellent future scope. As the development of smart cities continues to grow, the demand for accurate and efficient object detection algorithms for surveillance applications will only increase. Transfer learning can help address some of the challenges associated with training object detection models on limited data.

One future direction for integrating transfer learning with the YOLO model for smart city surveillance is the development of more advanced transfer learning techniques. For example, recent research has explored domain adaptation and meta-learning techniques to improve the transferability of pre-trained models to new domains. These techniques could be applied to the YOLO model for smart city surveillance to improve its accuracy and adaptability to new environments. Another future direction is the development of more specialized pre-trained models for smart city surveillance. For example, pre-trained models could be developed to detect objects of interest in different smart city environments, such as traffic intersections or pedestrian walkways. These specialized models could be fine-tuned on smaller datasets of images from these environments to improve their accuracy and efficiency. Overall, integrating transfer learning with the YOLO model for smart city surveillance has a bright future, with many opportunities for further research and development. Invading people's personal space is a major moral issue with public monitoring. When used for real-time object detection, the YOLO model may secretly collect and evaluate information about persons. Concerns concerning personal privacy and data security have been raised. When dealing with sensitive information, prolonged data storage might compromise privacy. People in public areas should be made aware of monitoring technologies and allowed to opt out whenever feasible. Trust must be preserved, and people's rights must be respected *via* open communication. As the demand for accurate and efficient object detection algorithms continues to grow, transfer learning techniques could help to drive progress in this area and enable the development of more effective surveillance systems for smart cities.

### Funding

This work was supported by Princess Nourah bint Abdulrahman University Researchers Supporting Project number (PNURSP2024R410), Princess Nourah bint Abdulrahman University, Riyadh, Saudi Arabia. The funders had no role in study design, data collection and analysis, decision to publish, or preparation of the manuscript.

### Grant Disclosures

The following grant information was disclosed by the authors:
Princess Nourah bint Abdulrahman University Researchers Supporting Project number: PNURSP2024R410.
Princess Nourah bint Abdulrahman University, Riyadh, Saudi Arabia.

### Competing Interests

The authors declare there are no competing interests.

### Author Contributions

- Surjeet Dalal conceived and designed the experiments, performed the experiments, analyzed the data, performed the computation work, prepared figures and/or tables, authored or reviewed drafts of the article, project supervision, and approved the final draft.
- Umesh Kumar Lilhore conceived and designed the experiments, performed the experiments, analyzed the data, performed the computation work, prepared figures and/or tables, authored or reviewed drafts of the article, and approved the final draft.
- Nidhi Sharma conceived and designed the experiments, performed the experiments, analyzed the data, performed the computation work, prepared figures and/or tables, authored or reviewed drafts of the article, methodology, and approved the final draft.
- Shakti Arora conceived and designed the experiments, performed the experiments, analyzed the data, performed the computation work, prepared figures and/or tables, authored or reviewed drafts of the article, collaboration, and approved the final draft.
- Sarita Simaiya conceived and designed the experiments, performed the experiments, analyzed the data, performed the computation work, prepared figures and/or tables, authored or reviewed drafts of the article, project review, and approved the final draft.
- Manel Ayadi conceived and designed the experiments, performed the experiments, analyzed the data, performed the computation work, prepared figures and/or tables, authored or reviewed drafts of the article, and approved the final draft.
- Nouf Abdullah Almujally conceived and designed the experiments, performed the experiments, analyzed the data, performed the computation work, prepared figures and/or tables, authored or reviewed drafts of the article, and approved the final draft.
- Amel Ksibi conceived and designed the experiments, performed the experiments, analyzed the data, performed the computation work, prepared figures and/or tables, authored or reviewed drafts of the article, and approved the final draft.

## Data Availability

The yolov7_dataset Computer Vision Project is available at https://universe.roboflow.com/yolov7-e9cjp/yolov7_dataset/dataset/1.

## Supplemental Information

Supplemental information for this article can be found online at http://dx.doi.org/10.7717/peerj-cs.1939#supplemental-information.

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
