# Peer review of "Improving smart home surveillance through YOLO model with transfer learning and quantization for enhanced accuracy and efficiency"

_PeerJ Computer Science, doi:10.7717/peerj-cs.1939_

## Round 0.1 · original submission · Major Revisions

As per comments from both reviewers, this paper can be considered with a major revision.

·

Basic reporting

The paper's title is clear and directly relates to the research topic, effectively communicating its primary focus. However, in the abstract, I recommend the addition of a sentence or two that highlights the primary contribution or novelty of the paper to provide readers with a more immediate understanding of the work's significance.

The keywords chosen are relevant and appropriate, aiding in the paper's discoverability.

In the introduction, there's a comprehensive background provided about smart cities and IoT security. Nevertheless, I noticed that the research questions aren't explicitly stated. The authors should consider formulating 2-3 distinct research questions or hypotheses. These would provide readers with a clearer sense of the study's direction.

Experimental design

Moving on to the methodology, there's a need for more detail. The authors should elaborate on how they calculated performance measuring parameters. Also, a description of any data preprocessing or manipulation steps would provide clarity.

Concerning the datasets, it would be beneficial to detail the source of these datasets, the nature of the images included, and the criteria used to split the data between training and testing. Given the choice of datasets from Roboflow, it would be pertinent to explain why these specific datasets were selected and how their unique characteristics validate the model proposed.

While the paper provides an overview of the YOLO method, there's notable repetition about its advantages and limitations. A more streamlined presentation of these points would enhance clarity. The proposed methodology section would benefit from a discussion on challenges faced during the integration of Transfer learning with YOLO. Furthermore, the mention of a 'bright future' should be substantiated with relevant trends or data.

As for the algorithmic approach, it would be enlightening to learn about any trade-offs encountered during algorithmic modifications and any anomalies observed during testing.

Validity of the findings

Turning to the findings, the results section would benefit from detailed metrics, including precision, F-measure, recall, and detection rate. The discussion should delve deeper into the implications of the results and potential applications in real-world scenarios.

It would also be worthwhile to compare the proposed model's effectiveness against 2-3 state-of-the-art models. Such a comparison would provide readers with a context regarding the model's performance.

When tables and figures are integrated, a detailed commentary on them would be essential. The claim about the model's immunity to environmental noise should be justified with specific experimental results or data.

Additional comments

Regarding the general presentation, the literature review could be more navigable if summarised by themes or categories. Some sections, especially the introduction, need to be more concise for better reader engagement. Furthermore, it's crucial to ensure that all tables and figures are correctly integrated, labelled, and referenced within the text.

Reviewer 2 ·

Basic reporting

This article delves into the challenges of analyzing anomalous behavior in crowd scenes using CCTV systems. Emphasizing the complexity of detecting violent behavior, the study predominantly focuses on deep learning techniques for action recognition and violence detection. The paper introduces an optimized YOLO model with transfer learning for surveillance in smart city home environments, achieving a 98.27% accuracy. The method promises enhanced ecological monitoring in smart cities.

Overall, the paper is well written and structured. However, it needs improvement on several aspects as described below. Taking into account these suggestions could provide a more comprehensive, insightful, and practically applicable research paper.

1. The paper covers a broad spectrum, from object identification to crowd analysis and violence detection. Narrowing the focus might provide deeper insights into a particular area, making the research more valuable for specific applications.

2. While the model is tested on 7382 images, it would be more robust if multiple diverse datasets, including real-world scenarios, were used. This would ensure the model's generalizability across various environments and not just the tested conditions.

3. Using the YOLO model with transfer learning and quantization could be computationally expensive. The paper should discuss computational costs, making it easier for potential users to assess feasibility.

Experimental design

4. The work mentions that the proposed method outperforms the conventional one. A clearer, side-by-side comparison with other popular models or techniques would offer readers a better perspective on the model's strengths and weaknesses.

5. The paper would benefit from a section discussing potential real-world challenges when implementing the model in smart city environments. Factors like varying light conditions, occlusions, and diverse human behaviors might affect the model's performance.

6. While quantization is a step forward, other techniques such as pruning or distillation might further optimize the model. These potential avenues can be explored to make the model even more efficient.

Validity of the findings

7. With surveillance and monitoring, there's always a concern regarding privacy. The paper should discuss the ethical implications of implementing such systems, especially in public spaces in smart cities.

8. The paper boasts a 98.27% accuracy, but it would be beneficial to understand the false positive and false negative rates. In security scenarios, false positives can be as problematic as false negatives.

9. Consider implementing a feedback mechanism. If the model makes an incorrect prediction, this mechanism would help the model learn from its mistake, making it adaptive over time.

---

## Round 0.2 · Minor Revisions

As per comments from original reviewers, a minor revision can be recommended.

·

Basic reporting

The authors have improved the manuscript's clarity, especially in the abstract by elucidating the novelty of their approach. The title is appropriately descriptive of the content, and the background provided sets a firm context for the research conducted. Nonetheless, there is still a need to provide more granular detail regarding the dataset used, particularly its size and diversity, to adhere to the PeerJ transparency policy.

Experimental design

The research design is solid, fitting well within the journal’s scope, and the methodological framework is sufficiently detailed to allow replication. The research question is meaningful and aligns with the identified knowledge gap. Ethical considerations are appropriately addressed.

Validity of the findings

The results are robust, statistically sound, and support the conclusions drawn. The study's outcomes are indeed linked to the original research questions, adding to its coherence. Nonetheless, a deeper explanation of the statistical methods used, especially in Section 4.2, would fortify the findings' validity. The discussion in Section 5 should be further refined to align with the research questions and hypotheses, ensuring a seamless narrative flow from introduction to conclusion.

Additional comments

The manuscript contributes an innovative perspective to smart home surveillance, with potential implications for future smart city monitoring systems. While the authors have made significant revisions, some sections would benefit from language and grammatical refinements to maintain professional clarity. The literature review in Section 2 should also be expanded to more explicitly address the study's contribution to closing the identified knowledge gap.

Reviewer 2 ·

Basic reporting

The authors have addressed all the reviewers' comments; no further revision is required.

Experimental design

This part has significantly been improved. Nothing to add.

Validity of the findings

This part is quite good. The validation on two datasets comprising 7382 images demonstrates the method's effectiveness.

---

## Round 0.3 · accepted · Accept

This revised paper can be accepted in the current version.

·

Basic reporting

The authors have addressed my concerns regarding dataset transparency and professional English usage. The additional details enhance the manuscript's clarity and adherence to the journal's standards.

Experimental design

I commend the authors for their thorough consideration of ethical standards and the clarity in describing their methodological approach. The improvements significantly enhance the manuscript's quality and relevance within the journal's scope.

Validity of the findings

The enhanced statistical explanations and discussion revisions directly align with the original research questions, thereby strengthening the validity of the findings. The authors have convincingly demonstrated the robustness and statistical soundness of their data.

Additional comments

The authors' efforts to refine the language and expand the literature review have notably improved the manuscript's comprehensiveness and professional quality.

Reviewer 2 ·

Basic reporting

The authors have addressed all the reviewers' comments; no further revision is required. The quality of the paper has significantly been improved.

Experimental design

This part has significantly improved after addressing the reviewers' comments.

Validity of the findings

Nothing to add in this part.

Additional comments

The paper is ready for publication.